## Protocol for an intervention development and pilot implementation evaluation study of an e-health solution to improve newborn care quality and survival in two low-resource settings, Malawi and Zimbabwe: Neotree

Emma Wilson ,[1] Hannah Gannon,[1] Gwendoline Chimhini,[2] Felicity Fitzgerald ,[3] Nushrat Khan,[1] Fabiana Lorencatto,[4] Erin Kesler,[5] Deliwe Nkhoma,[6] Tarisai Chiyaka,[7] Hassan Haghparast-Bidgoli ,[8] Monica Lakhanpaul ,[1] Mario Cortina Borja,[1] Alexander G. Stevenson,[9] Caroline Crehan,[1] Yali Sassoon,[10] Tim Hull-Bailey,[1] Kristina Curtis,[4] Msandeni Chiume,[11] Simbarashe Chimhuya,[12] Michelle Heys[1]

For numbered affiliations see end of article.

**Correspondence to**
Dr Michelle Heys;
m.heys@ucl.ac.uk

## ABSTRACT

**Introduction** Every year 2.4 million deaths occur worldwide in babies younger than 28 days. Approximately 70% of these deaths occur in low-resource settings because of failure to implement evidence-based interventions. Digital health technologies may offer an implementation solution. Since 2014, we have worked in Bangladesh, Malawi, Zimbabwe and the UK to develop and pilot Neotree: an android app with accompanying data visualisation, linkage and export. Its low-cost hardware and state-of-the-art software are used to improve bedside postnatal care and to provide insights into population health trends, to impact wider policy and practice.

**Methods and analysis** This is a mixed methods (1) intervention codevelopment and optimisation and (2) pilot implementation evaluation (including economic evaluation) study. Neotree will be implemented in two hospitals in Zimbabwe, and one in Malawi. Over the 2-year study period clinical and demographic newborn data will be collected via Neotree, in addition to behavioural science informed qualitative and quantitative implementation evaluation and measures of cost, newborn care quality and usability. Neotree clinical decision support algorithms will be optimised according to best available evidence and clinical validation studies.

**Ethics and dissemination** This is a Wellcome Trust funded project (215742_Z_19_Z). Research ethics approvals have been obtained: Malawi College of Medicine Research and Ethics Committee (P.01/20/2909; P.02/19/2613); UCL (17123/001, 6681/001, 5019/004); Medical Research Council Zimbabwe (MRCZ/A/2570), BRTI and JREC institutional review boards (AP155/2020; JREC/327/19), Sally Mugabe Hospital Ethics Committee (071119/64; 250418/48). Results will be disseminated via academic publications and public and policy engagement

## STRENGTHS AND LIMITATIONS OF THIS STUDY

⇒ Mixed methods intervention codevelopment and pilot implementation underpinned by behavioural science frameworks will optimise acceptability, feasibility and usability of Neotree, and the implementation strategy for larger scale roll out.
⇒ Piloting of quantitative and qualitative clinical, quality of care, process and economic measures will allow for a robust protocol for larger scale evaluation.
⇒ Collecting case fatality rate data from five hospitals in low-resource settings will ensure informed sample size calculations for larger scale evaluation.
⇒ Clinical, quality of care and demographic data from an anticipated 15 000 sick and vulnerable babies will enable studies of patterns, causes and risk factors for mortality and key diagnoses (eg, sepsis/neonatal encephalopathy).
⇒ Clinical and cost-effectiveness data will not be possible through this study design.

activities. In this study, the care for an estimated 15 000 babies across three sites will be impacted.
**Trial registration number** NCT0512707; Pre-results

## INTRODUCTION

Worldwide, 2.4 million children younger than 28 days die yearly, accounting for 48% of deaths in children under 5[1]. Approximately 70% of newborn deaths are avoidable through the implementation of simple, evidence-based interventions.[2] Health systems strengthening and training in newborn care are key to saving newborn lives.[3–5] Implementation of

BMJ

evidence-based guidelines can be supported through provision of reliable data systems, clinical decision support tools and education.[6 7] We are developing and robustly evaluating an integrated quality improvement system for hospital-based sick and vulnerable newborns: Neotree. Neotree combines evidence-based clinical guidelines with real-time newborn data collection, data visualisation and export and newborn education on one platform.[8] This tablet-based digital system is for use at the hospital bedside by healthcare professionals (HCP) with a range of skills and competencies (primarily nurses) supporting the care and treatment of newborns. Neotree development has followed standard software development and Medical Research Council (MRC) complex intervention development frameworks.[9]

### Background work to Neotree
A literature review of HCP-led newborn interventions in low-resource settings (LRS) identified gaps in successful implementation of proven interventions.[10] To address these gaps, we explored the concept and acceptability of digital data capture and clinical decision support via workshops with HCP in Bangladesh (n~15; 2014, unpublished) and developed a prototype app (2015). An editor platform was designed to allow a clinician to configure the data capture forms and Neotree-alpha was configured (2016). Next, a qualitative study was conducted with HCP in Zomba Central Hospital (ZCH), Malawi to understand barriers to delivering quality newborn care and to explore the potential for digital health interventions to mitigate these.[8] We selected ZCH as a Neotree co-investigator had previously piloted a clinical algorithm to support babies with respiratory distress at this site.[11] Neotree-beta minimal viable product 1 (MVP-1) was subsequently codeveloped with Malawian HCP (n=46, 2016–2017) in the same clinical setting.[8] Neotree-beta MVP-1 included data capture on admission, resuscitation clinical diagnostic and management support and associated educational material. Clinical management advice pages were designed and linked to HCP chosen diagnoses. Neotree was found to be acceptable, feasible and highly usable and the potential for electronic clinical audit data demonstrated.[8 12] HCP reported improved perceived ability to deliver quality care.[8] Neotree implementation did not continue at ZCH due to lack of ongoing funding. Identified strategies to optimise implementation included the introduction of a technical support role (suggested title: Neotree Ambassador).

In November 2018, Neotree was introduced at Sally Mugabe Central Hospital Neonatal Unit, Harare, Zimbabwe, at the request of local clinical teams,[13] presenting an opportunity to codevelop and test Neotree in a doctor-led unit within a new LRS with unique health system challenges. Neotree Ambassador roles were included in deployment, discharge data capture was developed, and linkage between clinical and laboratory blood culture data was undertaken to improve management of neonatal sepsis. A second site was identified in Malawi—Kamuzu Central Hospital—for further development and piloting. Since May 2019, we have been further developing Neotree-beta MVP-2 data capture on admission and discharge at this site, codeveloping a data

dashboard prototype and assessing usability, acceptability, barriers to implementation, usage and feasibility of both of these functions, using behavioural science frameworks. The data dashboard prototype included a summary statistics page, an admission hypothermia page and monthly morbidity mortality statistics developed using PowerBi.

Following this work, our priority has been the ongoing development and pilot implementation evaluation of remaining functions of Neotree, to create Neotree-gamma. Neotree-gamma will include data capture (admission, discharge and laboratory); clinical decision support (resuscitation and non-resuscitation diagnosis and management); education; and data transfer to local teams, dashboards and national databases. As HCP complete the admission, they will receive prompts to respond appropriately to the data they have entered and manage patients according to evidence-based guidelines. Neotree works offline to synchronise with a network when available. It is a not-for-profit venture with open-source code and can be preconfigured to adapt to available resources (medication and technology) within the index facility.

### Aims and objectives
Our primary research question is:
► *Can an integrated digital quality improvement system be implemented and sustained in two hospitals in Zimbabwe and one hospital in Malawi to improve newborn care?*

Secondary research questions include:
► *What is the case-fatality rate of intervention hospital neonatal units before and after implementation of Neotree?*
► *What is the case-fatality rate of two matched control hospital neonatal units not implementing Neotree during this time period?*
► *Can a predictive algorithm be developed to improve identification of neonatal sepsis?*
► *What is the sensitivity/specificity of Neotree sepsis algorithm versus gold standard?*

Study aims and objectives are to:
1. Further develop, implement and evaluate Neotree at three intervention sites.
2. Collect outcome data for newborns admitted to two hospitals where Neotree is not being implemented.
3. Test the clinical validity of Neotree neonatal sepsis diagnostic algorithm against gold standard diagnosis.
4. Add data visualisation and linkage to Neotree functionality.
5. Develop and test proof of concept for communicating daily electronic health records (EHR) using Neotree.

## METHODS AND ANALYSIS
### Study design
This is a mixed methods intervention codevelopment, pilot implementation and economic evaluation study (which will run from 7 October 2019 to 6 April 2022). We will continue to follow the MRC complex intervention development framework.[14] Table 1 shows anticipated timeline and study overview.

| Table 1 | Intervention roll-out and study timelines | |
|---|---|---|
| Months 1–4 | Study set up | Protocol development, recruitment and securing necessary permissions/approvals. |
| Months 1–21 | Ongoing embedding of Neotree into standard care and continuation of data collection in SMCH and KCH as part of studies (REF: P.02/19/2613) | |
| Months 5–24 | Implementation of Neotree into standard clinical care at CPH | |
| Months 1–9 (or until sample size reached) | Clinical validation substudy | Clinical validity of the diagnostic algorithms |
| Months 5–9 | Ongoing data dashboard development and data linkage | Codevelopment and optimisation the dashboard via design/usability workshops |
| Months 6–21 | Pilot implementation science evaluation | Qualitative studies with HCP (nurses, nursing students and doctors), hospital administration staff (senior doctors/nurses, managers), and parents/carers |
| Months 6–21 | Collection of quality of newborn care measures | |
| Months 10–15 | Testing concept of electronic record system | Configure and test linkage of Neotree data to the MoH electronic record system |
| Months 1–21 | Economic evaluation/costings data | During all phases, cost and resource implications data will be collected and analysed. |
| Months 21–24 | Data analysis and write up | |
| HCP, healthcare professionals. | | |

## Setting

This is a two-country study in Malawi and Zimbabwe where, in 2019, the neonatal mortality rates were 19.7 and 26.2 per 1000 births, respectively.[15] In Zimbabwe, Neotree will continue to be implemented at Sally Mugabe Central Hospital, the largest of three newborn tertiary care facilities in Zimbabwe, delivering 12 000 newborns annually, where the 100-cot nursery often runs at 120%–130% capacity. Audit data show case fatality rates of 210 deaths per 1000 admissions.[13] Neotree will also be introduced to clinical processes at Chinhoyi Provincial Hospital, Zimbabwe—a provincial level hospital, delivering 4500 newborns annually with audit data showing case fatality rates of 180 per 1000 admitted babies. Sally Mugabe Central Hospital is one of six central hospitals in Zimbabwe and delivers primarily doctor-led care. Chinhoyi Provincial Hospital is one of eight provincial hospitals in Zimbabwe and delivers predominantly nurse-led newborn care. In Malawi, Neotree will continue to be delivered at Kamuzu Central Hospital, one of four central hospitals in Malawi, delivering 4500 newborns annually. The neonatal unit admits around 2600 babies per year (from both within and outside of the hospital) with a case fatality rate of 210 per 1000 admitted babies.

## Study participants

Study participants include frontline and managerial staff involved in the delivery of newborn care (eg, nurses, doctors, and nursing students) at the three implementation sites, and other key personnel and stakeholders such as hospital managers and parents/caregivers of newborns admitted to newborn care units. We estimate that we will recruit ~170 HCP and 30 parents/caregivers across three sites in a phased pilot evaluation of each new codeveloped functionality of the Neotree system.

Routine clinical admission, discharge and microbiological data will be prospectively recorded on the Neotree system for all newborns admitted to newborn care units at the three intervention sites (~12 000 admitted babies).

## Consent procedures

Informed written consent will be sought from participants who take part in semi-structured interviews and focus group discussions. Participants with low literacy levels may give verbal consent which will be audio recorded and witnessed. We will follow international and local precedent for collection of neonatal pseudonymised data for the purposes of epidemiological surveillance and service evaluation such as the neonatal UK/Australia/New Zealand Badger net system,[16] and the WHO-led District Health Information Software (DHISv2).[17] Hence, we will not obtain informed consent from guardians to enter patient level data. No novel data will be recorded beyond that usually documented for clinical management in these settings, and no new procedures performed.

## Codevelopment and optimisation of the Neotree system

Data capture functions will be optimised and validated; data dashboards and clinical decision support functions will be fully developed; data linkage to Zimbabwean

national EHR and aggregate data DHISv2 systems demonstrated; and daily EHR functionality will be piloted.

## Data capture

A neonatal data dictionary will be developed in line with similar existing guidelines.[18 19] It will define the types and formats of data captured in Neotree, including standard data definitions to make it accessible to all data users. This will be made publicly available alongside open source code on GitHub (https://github.com/Neotree/Neotree). Data capture forms for admission, discharge and laboratory will be refined according to usability feedback. Data quality will be reviewed monthly to assure completeness and consistency of data by monitoring and reducing missing data where applicable. Annual prospective paper audits comparing Neotree capture of admissions and discharges to those recorded by ward paper records will be conducted. The data pipeline will be optimised to ensure automated data processing aligned with clinicians' needs and secure backup in UCL research databases.

## Dashboard development

The prototype initially developed at Kamuzu Central Hospital will be further developed in design/usability workshops with end-users. In both countries, one-to-one 'in vitro' user-tests will be conducted using a think aloud approach and an adapted user experience topic guide. During these sessions, pages of the dashboard (MVP-1) will be presented to HCP to gauge understanding and interpretation of the visualisations and collect feedback. Usability themes will be generated and analysed using agile rapid analysis. Following one-to-one user tests, participants will be invited to attend a videoed group workshop to consider new visualisations for MVP-2-dashboard. Optimal dashboard software will be identified.

## Clinical decision support

Clinical decision support will be refined according to best available evidence and operationalised within the system. With Neotree, we will have a context-specific, detailed stream of clinical data at admission and outcome (discharge/death), in combination with laboratory diagnostic data for neonatal sepsis. With these data, we will build models to predict which babies are likely to benefit from specific interventions such as antibiotics. Clinical validation of the sepsis algorithm will be conducted retrospectively using data collected from Sally Mugabe and Kamuzu Central Hospitals. Admission diagnoses from the admitting healthcare professional, the senior clinician (based on admission data alone) and blood culture results (gold standard) will be retrospectively compared with Neotree algorithm diagnosis. Assuming sensitivity and specificity of 92% (lower 95% CI: 84%) >222 babies would need to be diagnosed with blood culture positive sepsis over 5 months, during which more than 2000 babies will be admitted with sepsis across sites.[20]

## Data linkage

We will develop and demonstrate the ability for Neotree data to be exported to the Zimbabwean national EHR system and to DHISv2, the most commonly used aggregate data system for reporting of health service data in the African region.

## Daily electronic health record

A scoping study of the potential to extend Neotree-gamma to include daily EHR will be conducted (n~20 babies).

## Implementation evaluation

We will conduct qualitative studies to assess the acceptability and feasibility of the functionalities of Neotree, informed by behavioural science theories and frameworks. Qualitative and quantitative usability data will be gathered in addition to quantitative measures of usage, patterns in clinical outcomes and measures of quality newborn care.

## Acceptability, feasibility and usability

Focus groups and individual interviews will be conducted with HCP, after each new functionality becomes embedded within clinical practice at three timepoints: pre-implementation (Chinhoyi Provincial Hospital only), implementation and sustainability.

In the pre-implementation phase (at baseline), we will explore current practice, quality improvement needs, and potential barriers and enablers to implementation at the new site of Chinhoyi Provincial Hospital, Zimbabwe. We will hold one focus group discussion with HCP (n~10) and approximately 10 semi-structured interviews with senior doctors, nurses and hospital administrators.

In the implementation phase, we will deploy Neotree to Chinhoyi Provincial Hospital while continuing to test and codevelop new functionalities across all sites. We will conduct 3 rounds of focus group discussions at each site (nine in total) with approximately 10 participants in each (n~90). Perceived acceptability, feasibility and usability will be explored as follows:

▶ Round 1: basic functionality of Neotree-beta MVP-2 (data capture at the bedside, clinical decision support for resuscitation and stabilisation and education)—Sally Mugabe Central and Chinhoyi Provincial Hospitals only.
▶ Round 2: data dashboards (all sites).
▶ Round 3: non-resuscitation clinical decision support (all sites).

To complement these data, we will conduct one set of individual interviews with approximately five senior clinical and managerial staff at each site (n~15). Topic guides for both focus groups and individual interviews will be semi-structured. Questions to explore acceptability of Neotree will be based on the domains from the Theoretical Framework of Intervention Acceptability,[21] for example, burden, intervention coherence, opportunity costs, ethicality. Questions to explore barriers

and enablers to implementing Neotree in practice will be based on the Theoretical Domains Framework[22]; an integrative framework of 33 behaviour change theories proposing 14 domains of factors facilitating/hindering behaviour change, for example, knowledge, available resources, social influences, motivation, and so on. Drafts of topic guides (online supplemental files 1–3) will be reviewed and piloted before the final versions are implemented. Focus groups and interviews will last for a maximum of two hours and one hour, respectively. These will be conducted by a trained researcher, either face-to-face in a private location in the hospital, or remotely via platforms such as Microsoft Teams, at a convenient date and time to participants.

Focus groups and interviews will be audio recorded, transcribed verbatim and fully anonymised. Anonymised data will be stored securely at UCL for 10 years. We will analyse the transcripts using a combined deductive framework and inductive thematic analytical approach,[23] to identify which domains are key influences on implementation and acceptability. To ensure reliability, a subset of 10% of transcripts will be double coded by another researcher (KC). We will compare themes over time (i.e., across the implementation period), across countries, and according to professional role. Refreshments will be provided to those taking part and travel costs will be reimbursed where relevant. Following analyses of focus group and interview data, we will identify any potential refinements or additions to be made to Neotree or the associated training materials, in order to improve acceptability, usability and address barriers and enablers to implementation. We will draw on behavioural science intervention development frameworks[24 25] to identify relevant behaviour change techniques to address identified barriers and enablers.

In the sustainability phase, we will conduct a final round of data collection with ~10 healthcare professionals and ~5 senior clinical and managerial staff at each site (total n~45) with a focus on intervention sustainability once research and software development teams have withdrawn. No further changes to functionality will be made to Neotree-gamma during this period. We will compare themes from sustainability vs initial implementation data collection period, across countries and across professional roles using the same methodology described above.

We will interview ~10 parents/carers of newborns at each of the three hospitals (~30 parents/carers) to explore their views on the use of digital innovations in healthcare, and the perceived acceptability of Neotree. Interviews will be semi-structured, based on the Theoretical Framework of Acceptability.[21] Interviews will be conducted by an independent researcher, who is not involved in the provision of clinical care. Assurances will be given to parents/caregivers that their participation will not affect the care of the baby or other family members. Findings will inform refinements of Neotree system and

associated training to ensure it is acceptable both to those providing and receiving newborn care.

### Analysis of routine health data

Admission data from intervention sites will be analysed monthly to estimate measures such as overall and disease specific case-fatality rate, quality of newborn care and usage. We will test the feasibility of collecting the quality of newborn care endpoints shown in Box 1, to inform the data collection procedures in any future large-scale evaluation. Most map directly to WHO standards of quality of maternal and newborn care in health facilities.[5]

---

**Box 1  Quality of newborn care endpoints**

Temperature on admission
1. The proportion of all newborns who had a normal body temperature (36.5–37.5°C) at the first complete examination (60–120 min after birth) (WHO quality statement 1.1b).

Documentation
2. The proportion of all newborns for whom there is documented information on (a) body temperature, (b) respiratory rate, (c) HR, (d) $O_2$ saturations in air, (e) saturations in oxygen, (f) blood sugar, feeding behaviour, (g) absence or presence of danger signs, (h) admission weight (WHO quality statement 1.1c)

Resuscitation
3. The proportion of all newborns who were not breathing spontaneously after additional stimulation at the health facility who were resuscitated with a bag-and-mask. (WHO quality statement 1.5)
4. The proportion of all newborns who were not breathing spontaneously after additional stimulation at the health facility who were resuscitated with a bag-and-mask within 1 min min of birth. (WHO quality statement 1.5)

Infection
5. The proportion of all newborns in the health facility *with signs of infection* who received injectable antibiotics. (WHO quality statement 1.7b)
6. The proportion of all newborns *of mothers with signs of infection* in the health facility who received injectable antibiotics. (WHO quality statement 1.7b)
7. Proportion of newborns *with suspected severe bacterial infection* who received appropriate antibiotic therapy. (WHO quality statement 1.8)
8. Proportion of newborns born to HIV +ve mothers who received Nevirapine on first day of life (discharge)
9. Proportion of neonates with low blood sugar who are treated with a feed or dextrose as appropriate.

Data collection and health system
10. The proportion of all newborns currently in the health facility who have a patient identifier and individual clinical medical record*. (WHO quality statement 2.1)
11. The proportion of all newborns discharged from the health facility within the past 24 h hours who had an accurately completed record of processes of care, treatments, outcomes and *diagnoses* (with ICD code). (WHO quality statement 2.1)
12. Data are collected routinely in the health facility during labour, childbirth and the postnatal period (and used regularly to make decisions on quality improvement)
13. The proportion of newborns seen in the health facility in the past 3 months who fulfilled the facility's criteria for referral who were actually referred. (WHO quality standard 3.1)

---

## Economic evaluation-intervention costing

During all phases, cost and resource implication data will be collected and analysed to determine the costs of developing and piloting Neotree from the provider perspective. These include Neotree development costs, training, planning and set up, implementation, and resource implications for the hospital/healthcare system. Costs of developing and implementing the Neotree system will be collected through expenditure reports, time-use surveys and interviews with project staff (online supplemental files 4 and 5). Information on potential impact on intervention hospitals will be collected through time-use surveys with all HCP involved in Neotree development and implementation, supplemented with project records on their involvement in different intervention activities, such as software development and training workshops (i.e., opportunity costs). To measure the effect of implementation on time spent on procedures/activities in the delivery of newborn care pathways, a pilot time-use survey will be conducted (implementation phase) at all intervention sites and one comparable site in each setting, to record admission and discharge activities and time spent on each activity for around 10 newborn patients in each hospital (n=30 in total). The tools developed and used for economic data collection will be modified and adopted for future use in a larger trial/evaluation study of Neotree.

## Protocol for future large-scale evaluation

Data from all phases will inform the study protocol, costings and implementation strategy for a future large-scale roll-out and evaluation.

To inform the sample size calculation for a large-scale evaluation (e.g., a stepped wedge trial), we will collect 6 months of clinical, morbidity and mortality data from newborns admitted to two comparator hospitals providing usual care in Zimbabwe. These are: Mbuya Nehanda Maternity Hospital (part of the Parirenyatwa hospital group) and Bindura Provincial Hospital. Selection of comparator hospitals was based on geographical proximity and similarities in catchment area and service-level provision to intervention hospitals. Data from paper-based admission forms will be entered retrospectively at these sites onto the Neotree app. In Malawi, aggregate electronic routine case fatality data from representative health facilities will be attained from the Health Management Information System (i.e., from a central hospital, with nurse-led provision and a similar catchment area to Kamuzu Central Hospital).

Of note, data collected during this pilot implementation phase are insufficient and not designed to test clinical and cost effectiveness—rather they will be used to refine Neotree and to inform a robust evaluation. Primary, secondary, process, quality of care and economic outcomes will be clarified, alongside data collection procedures and our logic model.

## Patient and public involvement

No patients were directly involved in the design of the study. However, this protocol responds to findings from our pilot acceptability and feasibility work in Malawi in 2016, which indicated that HCP were concerned about parent perceptions of the use of Neotree in routine newborn care.[8] They reported that some parents were 'afraid they think you are taking the information someplace else'. Student-nurses reported that parents/guardians thought the tablets were a distraction and that 'we are just on social networks', while others viewed it positively 'with gladness'.[8 26]

In response, this study will explore the acceptability of Neotree via semi-structured interviews with parents/carers across the three implementation sites, to capture patient perspectives. We will also develop a patient and public involvement strategy. This will include working with partners (Art & Global Health Center Africa and the UCL Co-Production Collective) to build the capacity of mothers/carers so that they may support dissemination of study results in catchment areas of intervention hospitals, and coproduce future iterations of Neotree. We will use participatory methods, such as community dialogues, to disseminate study results.

## Data management plan

Study protocols, training manuals and data collection tools will be made available on a study website. On study completion, all documents and record forms will be stored onsite for at least 10 years. Our team has had extensive discussions with clinical teams and Ministries of Health in setting up this study. Our agreed data management plan for newborn data is in online supplemental file 6. A subset of the anonymised research database will be made open source after publication of the main study findings to ensure maximum reach and accessibility.

Focus groups, semi-structured interviews and workshops will be recorded on two digital audio recording devices. Recordings will be anonymised and transcribed verbatim using password-protected laptops. Hard copies of field notes and transcripts will be anonymised and locked away. Soft anonymised copies will be stored on a secure laptop. Similarly, data collected through time-use surveys will be anonymised and stored on a secure laptop.

## ETHICS AND DISSEMINATION

Research ethics approvals for this study were granted from: Malawi College of Medicine Research and Ethics Committee (P.01/20/2909; P.02/19/2613); University College London (17123/001; 6681/001; 17123/001; 5019/004); Medical Research Council Zimbabwe (MRCZ/A/2570); Biomedical Research Training Institute institutional review board, Zimbabwe (AP155/2020), Joint Research Ethics Committee, University of Zimbabwe review board (JREC/327/19), and Sally Mugabe Hospital Ethics Committee (HCHEC 250418/48; 071119/64).

## Potential ethical considerations

There are no conflicts of interest. No new drugs/biologic agents will be administered to the participants during this

study, nor will previously used agents be used in a new manner. No additional tests of clinical management will be initiated beyond those of standard practice. Neotree will support the consistent implementation of national/international evidence-based guidelines.

## Dissemination

Our output management plan—codeveloped with our partners and co-investigators—will be reviewed every 3 months. Discussions with the Zimbabwean Ministry of Health are ongoing about how the Neotree could, if successful, be rolled out across the country. We will develop long-term data sharing and access procedures (including a Data Access Committee with independent academic and lay members to assess requests for aggregated data).

A stakeholder event (Zimbabwe) will focus on synergies in research needs between Malawi and Zimbabwe, using Neotree as a platform for African-driven research questions, projects and higher degrees. It will include presentations on quality improvement projects/audits completed using Neotree. An anonymised research database comprising clinical newborn data from all three hospital sites will be designed at the Neotree collaborator meeting with a data management plan setting out conditions for access according to national/international guidelines. We will disseminate our discussions/conclusions in an opinion piece in a peer-reviewed journal.

## Impact of COVID-19

Our research programme has continued throughout the COVID-19 pandemic.[27] Key changes to our research plan are described in online supplemental file 7.

## DISCUSSION

Few digital interventions in LRS have been well described and rigorously evaluated thus limiting their potential with respect to clinical and implementation effectiveness, sustainability and generalisability. The work described here will allow the completion of codevelopment of Neotree-gamma with key functionalities configured, operationalised, tested and ready for larger scale roll out and evaluation across Zimbabwe and Malawi. Behavioural science theory and frameworks will be used to explore barriers/enablers to implementation and inform intervention refinement and development of strategies to encourage implementation at scale and long term sustainability.[21–23] The economic evaluation will estimate pilot implementation costs and resource requirements for sustainability and scale up, and affordability. Clinical outcome and cost data will inform sample size and potential resources for a large-scale evaluated roll out. It should be acknowledged that the costs and resource use at the pilot hospitals will not necessarily be generalisable to other hospitals, but we hope that the detailed cost analysis alongside the pilot, will provide information to estimate potential costs at scale.

Our overall vision is to use evidenced-based best practice and information technology to improve clinical decisions for newborn care and increase rates of newborn survival in under-resourced healthcare settings. We aim to create a locally led and curated database of aggregate newborn outcomes that can be used to robustly evaluate health systems, undertake healthcare planning and resource allocation at both the microlevel (by site), regional, national and international levels. In addition, we are committed to open-source code ensuring the Neotree code is freely available for countries and hospitals to adopt, modify and run, and enabling those hospitals and ministries to own and control their data.

This work addresses a primary sustainable development goal—reducing neonatal mortality in LRS—and addresses key strategic aims of the Every Newborn Action Plan.[28] Our clinical algorithms and linkage to microbiology data aim to optimise the management of leading causes of newborn death. In this study, the care for an estimated 15 000 babies across the three test sites will be impacted by Neotree. Through successful rollout across Zimbabwe and Malawi—the care for nearly 300 000 babies could be improved annually.

**Author affiliations**
[1]Population, Policy and Practice Research and Teaching Department, UCL Great Ormond Street Institute of Child Health, London, UK
[2]Unit of Child and Adolescent Health, Faculty of Medicine and Health Science, University of Zimbabwe, Harare, Zimbabwe
[3]Infection, Immunity and Inflammation Research & Teaching Department, UCL Great Ormond Street Institute of Child Health, London, UK
[4]UCL Centre for Behaviour Change, London, UK
[5]The Children's Hospital of Philadelphia, Philadelphia, Pennsylvania, USA
[6]Parent and Child Health Initiative Trust, Lilongwe, Central Region, Malawi
[7]Biomedical Research and Training Institute, Harare, Zimbabwe
[8]UCL Institute for Global Health, London, UK
[9]Mbuya Nehanda Maternity Hospital, Harare, Zimbabwe
[10]Snowplow Analytics, London, UK
[11]Department of Paediatrics, Kamuzu Central Hospital, Lilongwe, Malawi
[12]Unit of Child and Adolescent Health, Faculty of Medicine and Health Sciences, University of Zimbabwe, Harare, Zimbabwe

**Collaborators** Collaborator Group Name: Neotree team Dr Michelle Heys (UK), Professor Mario Cortina Borja (UK), Professor Monica Lakhanpaul (UK), Dr Fabiana Lorencatto (UK), Dr Hassan Haghparast-Bidgoli (UK), Tim Hull-Bailey (UK), Samuel Neal (UK), Edna Mugwagwa (UK), Yali Sassoon (UK), Daniel Silksmith (UK), Nushrat Khan (UK), Dr Felicity Fitzgerald (UK), Dr Caroline Crehan (UK), Dr Emma Wilson (UK), Erin Kesler (USA), Dr Hannah Gannon (Zimbabwe - UK), Dr Gwendoline Chimhini (Zimbabwe), Dr Simbarashe Chimhuya (Zimbabwe), Tarisai Chiyaka (Zimbabwe), Dr Msandeni Chiume (Malawi), Deliwe Nkhoma (Malawi), Dr Yamikani Mgusha (Malawi), Danie Krige (South Africa), Louis Du Toit (South Africa), Morris Baradza (Zimbabwe), Farai Matare (SA - Zimbabwe).

**Contributors** MH and EW wrote the first draft of the protocol paper with all other co-authors inputting to content and approving the final draft. MH, EW and HG led the writing of the research ethics applications. MH designed the study methods with guidance and input from all co-authors. Co-authors also offered specific input and expertise as follows: FL, EW and KC on behavioural sciences methodology; FF, HG, AS, SC, GC, CC, DN, TC, THB, EK and MCB on clinical validation and implementation; MCB on quantitative analyses; ML on parent/carer qualitative study and public engagement; CC on usability testing, data dashboard development and assessment of impact on quality newborn care; YS, THB and NK on software development and data pipeline; and HHB on economic evaluation.

**Funding** This work was supported by Wellcome Trust grant number 215742_Z_19_Z.

**Competing interests** MH, YS, EK and FF are trustees of the Neotree charity (www.neotree.org) but receive no financial payment from this role. CC was a trustee of the Neotree charity until 2018 and received no financial payment for this role.

**Patient and public involvement** Patients and/or the public were not involved in the design, or conduct, or reporting, or dissemination plans of this research.

**Patient consent for publication** Not applicable.

**Provenance and peer review** Not commissioned; externally peer reviewed.

**ORCID iDs**
Emma Wilson http://orcid.org/0000-0001-7091-2417
Felicity Fitzgerald http://orcid.org/0000-0001-9594-3228
Hassan Haghparast-Bidgoli http://orcid.org/0000-0001-6365-2944
Monica Lakhanpaul http://orcid.org/0000-0002-9855-2043

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
