## [Reviewer comments · BMJ Open]

ARTICLE DETAILS

TITLE (PROVISIONAL)	Protocol for an intervention development and pilot implementation evaluation study of an e-health solution to improve newborn care quality and survival in two low resource settings, Malawi and Zimbabwe: Neotree.
AUTHORS	Wilson, Emma; Gannon, Hannah; Chimhini, Gwen; Fitzgerald, Felicity; Khan, Nushrat; Lorencatto, Fabiana; Kesler, Erin; Nkhoma, Deliwe; Chiyaka, Tarisai; Haghparast-Bidgoli, Hassan; Lakhanpaul, Monica; Cortina Borja, Mario; Stevenson, Alexander; Crehan, Caroline; Sassoon, Yali; Hull-Bailey, Tim; Curtis, Kristina; Chume, Msandeni; Chimhuya, Simbarashe; Heys, Michelle

VERSION 1 – REVIEW

REVIEWER	Ganapathy, Krishnan Apollo TeleHealth Services, Telemedicine
REVIEW RETURNED	19-Aug-2021

GENERAL COMMENTS	Observations by Reviewer on Article bmjopen-2021-056605 ----- evaluation study of an e-health solution to improve newborn care quality and survival The authors have submitted a study protocol for a planned ongoing research study. The study design and methodology appears to have factored in all relevant factors and the authors are to be congratulated for attention to minute details. Actual execution and implementation of the protocol in the real world offers several challenges particularly in a Low Resource Setting environment . Training, retraining, learning, unlearning, relearning, trouble shooting, internet, WiFi, power back up, alternative strategies, motivation, incentivisation, spelling out job profile of all those involved in study, taking care of possible staff attrition, including redundancy not only hardware and software but HR as well. Competency of end users involved in the actual implementation and utilisation of the new innovative service, reproducibility after the trial is over digital literacy, general literacy, experience. The protocol should be culture sensitive taking into account factors mentioned above. Ensuring continuity of all those involved in the study is critical. At the same time after this study is over one should be able to have it integrated into the health care delivery system The reviewer however has some concerns which are enumerated below.  1. Actual time taken to initially learn and subsequently to actually use this software should be measured on Day -1, Day 0, day 1-3 , 6 weeks, 3 months etc . There would be an inherent resistance to any additional work both in times of effort and time 2. Extrapolating observations obtained from a hyperlocal environment to inferences and extending them to other regions could be a challenge .
--

	3. For comparison with controls, 6 months newborn clinical outcome data from 2 hospitals in Zimbabwe (why has a control group in Malawi not being considered is there some published data which shows that neonatal management is the same in both countries ??) providing usual care without additional eHealth solutions are being studied. Authors need to elaborate on what exactly is meant by retrospective . The control group data should ideally be from concurrent case records filled at the same time when the study is being done ,from 2 OTHER HOSPITALS without this intervention 4. Control data where there is no additional ehealth assistance, is restricted to 10% of the data that could be available. Reviewer why would like to know how this figure was arrived at and what statistical measurement was used. As this is only an analysis of data routinely entered in medical records ,in a busy non academic unit the chances of some information not being available cannot be excluded. Incomplete data in the control comparative group will have to be robust . To directly attribute the differences in outcome in the control group and the intervention group exclusively to the use of Neo True pe supposes that all confounding factors should be comparable and statistically significant. It is suggested that the control group should be much higher 5. It has been stated that “ ---- periodic prospective paper audits comparing NeoTree capture of admissions and discharges to those recorded by ward paper records will be conducted – to elaborate on how periodic is periodic ? 6. It Hs been stated that “ ----Six months of comparative newborn outcome data and cost data will be collected from two hospitals receiving usual care “. Where are these two hospitals located – Are the socio economic levels and standards of similar ? 7. Numbers have been mentioned throughout the submission . It would be useful yo know how these numbers wee chosen. Has it been computed by a biostatistician Is it taking into account confidence values, power etc if so this may be elaborated. If it is a convenience sampling taking into account the acual difficulties encountered in the real world this needs to be stated and a para or 2 added to assure the reader that these limitations would not significantly change interpretation 8. Proving cost effectiveness would go a long way in adding to the value of the proposed eHealth solution. The reviewer would be interested in reviewing the details of how this is proposed to be done – CAPEX, OPEX, maintenance, upgrades, connectivity time translating training time and deployment time into costs, tangible intangible costs . At the same computing the reduction in cost due to reducing mortality and morbidity will justify large scale deployment Developing eHealth solutions based on scientific evidence, and deploying it in real time in the real world, particularly in Low Resource Settings has the potential to significantly reduce morbidity and mortality in the newborn. Protocols for such studies need to factor in and give correct weightage to all confounding factors, so that even the sceptic and unbeliever will have to concede that it was an Android smart phone App which actually made all the difference!! The final result does not matter. It is the sincere attempt and willingness to consider multiple suggestions and be prepared to make mid course corrections if necessary that is more important K. Ganapathy Hon Distinguished Professor, The Tamilnadu Dr MGR Medical University Past President, Telemedicine Society of India & Neurological
--	---

	Society of India WHO Digital Health Expert
--	---

REVIEWER	Moradi-Lakeh, Maziar Iran University of Medical Sciences
REVIEW RETURNED	28-Nov-2021

GENERAL COMMENTS	 - Background characteristics of newborns admitted to the hospitals might be different by time and location, especially if the interventions, directly or indirectly, affect awareness of families to bring newborns sooner to the hospitals. I think it is important to clarify whether there is any measure to assess characteristics of the input neonates to the system? - Please provide data on baseline case fatality for the control hospitals - Please provide the link to access open-source code on GitHub - Please clarify whether a linkage is available between the NeoTree data and Malawi health system (similar to the linkage to the Zimbabwean health system) - The researchers have considered some interviews with parents/carers; are they involved in any of the processes which are related to NeoTree (such as data collection, visualization or export of data)? - If the process of NeoTree development is not repeated in each new setting, then its introduction costs to new hospitals will be different from the first few pilot hospitals. I think this should be clarified in the costing method, or should be considered as one of the limitations of costing - The supplementary tables have been slitted in several pages, and it's hard to follow the contents. I suggest to provide an excel version of facilitate reviewing and working with them.
--

VERSION 1 – AUTHOR RESPONSE

Reviewer: 1

The reviewer however has some concerns which are enumerated below.

1. Actual time taken to initially learn and subsequently to actually use this software should be measured on Day -1, Day 0, day 1-3 , 6 weeks, 3 months etc. There would be an inherent resistance to any additional work both in times of effort and time

-Thank you. Time taken to attend trainings and time taken to admit and discharge babies at intervention and comparator sites will be collected and will inform the economic evaluation. This is outlined on p13 of the manuscript.

-The quality of the data (e.g., data completeness) collected via the app will be monitored and barriers to using and engaging with Neotree will be explored via qualitative focus groups and interviews with

end users (p. 13).

2. Extrapolating observations obtained from a hyperlocal environment to inferences and extending them to other regions could be a challenge.

-We agree that determining the generalisability of findings is always a challenge. However, we are confident that through this pilot implementation study, involving three sites across two countries, we will be in a position to develop some preliminary hypotheses for how the intervention might work to improve quality of care; and how contextual factors may influence both 1) intervention effectiveness and 2) implementation effectiveness (how well the intervention is implemented in real world contexts). These hypotheses can then be tested in a future large scale roll out and evaluation.

3. For comparison with controls, 6 months newborn clinical outcome data from 2 hospitals in Zimbabwe (why has a control group in Malawi not being considered is there some published data which shows that neonatal management is the same in both countries??) providing usual care without additional eHealth solutions are being studied. Authors need to elaborate on what exactly is meant by retrospective. The control group data should ideally be from concurrent case records filled at the same time when the study is being done ,from 2 OTHER HOSPITALS without this intervention

- We have clarified that we are collecting data from comparator sites from Malawi as well (p14) via the Health Management Information System.

- We have elaborated that admission and discharge data is collected via paper-based forms at comparator sites in Zimbabwe. We will therefore employ a research assistant to enter data retrospectively into the Neotree app. We have clarified this on p 14.

4. Control data where there is no additional ehealth assistance, is restricted to 10% of the data that could be available. Reviewer why would like to know how this figure was arrived at and what statistical measurement was used. As this is only an analysis of data routinely entered in medical records ,in a busy non academic unit the chances of some information not being available cannot be excluded. Incomplete data in the control comparative group will have to be robust . To directly attribute the differences in outcome in the control group and the intervention group exclusively to the use of Neo True pe supposes that all confounding factors should be comparable and statistically significant. It is suggested that the control group should be much higher.

-Thank you. We have clarified on pages 13-14 that data from comparator sites is to inform the sample size calculation for a future large-scale evaluation (e.g., a stepped wedge trial). These data are not intended to test the potential clinical and cost effectiveness of the intervention. We will therefore only collect 6 months of data on neonates admitted to comparator sites to allow us to get a reasonable estimate of case fatality rates at these sites in order to inform our sample size calculation.

5. It has been stated that “ ---- periodic prospective paper audits comparing Neotree capture of admissions and discharges to those recorded by ward paper records will be conducted – to elaborate on how periodic is periodic ?

-Thank you, we have clarified that ‘periodic audits’ will take place annually. (p9 of the manuscript).

6. It has been stated that “ ----Six months of comparative newborn outcome data and cost data will be collected from two hospitals receiving usual care “. Where are these two hospitals located – Are the socio economic levels and standards of similar?

-We have clarified that these hospitals have similar catchment areas and models of care (i.e doctor led or nurse led). (p13-14)

7. Numbers have been mentioned throughout the submission. It would be useful to know how these numbers were chosen. Has it been computed by a biostatistician? Is it taking into account confidence values, power etc? If so, this may be elaborated. If it is a convenience sampling taking into account the actual difficulties encountered in the real world this needs to be stated and a para or 2 added to assure the reader that these limitations would not significantly change interpretation.

-Thank you. We have included one power calculation in the study based on accepted methods to compare a single binomial proportion with a hypothesised standard. This is to enable us to compare the sensitivity and specificity of the Neotree sepsis algorithm with gold standard blood culture results, and with HCP clinical diagnoses. We have cited the methodology for this sample size calculation in the protocol (p10).

-Our power calculation has been checked by the co-author Prof Cortina-Borja who is a Professor of Biostatistics.

-All other numbers cited in the protocol such as admission numbers and case fatality rates have been obtained from lead investigators at intervention facilities and are to provide readers with context for the clinical settings in which we will operate, rather than to make any inferences at this stage.

-Sample sizes for the qualitative studies are based on standard numbers of participants required for focus group discussions (8-10 per FGD), and interview studies (15-20 participants).

8. Proving cost-effectiveness would go a long way in adding to the value of the proposed eHealth solution. The reviewer would be interested in reviewing the details of how this is proposed to be done – CAPEX, OPEX, maintenance, upgrades, connectivity time translating training time and deployment time into costs, tangible intangible costs . At the same computing the reduction in cost due to reducing mortality and morbidity will justify large scale deployment

Developing eHealth solutions based on scientific evidence, and deploying it in real time in the real world, particularly in Low Resource Settings has the potential to significantly reduce morbidity and mortality in the newborn. Protocols for such studies need to factor in and give correct weightage to all confounding factors, so that even the sceptic and unbeliever will have to concede that it was an Android smart phone App which actually made all the difference!! The final result does not matter. It is the sincere attempt and willingness to consider multiple suggestions and be prepared to make mid course corrections if necessary that is more important.

-We agree with the reviewer that conducting economic evaluation alongside the pilot implementation will add value, in particular, where scarce evidence exist on cost and cost-effectiveness of e-health strategies such as Neotree, limiting their scale up. We aim to estimate costs of developing, setting up/introduction, and implementing Neotree in the pilot hospitals, from a provider perspective. We are conducting an economic costing, identifying and measuring all costs and resource use, including opportunity costs (e.g. training time, hospital staff involvement in development and operation etc) during the pilot period. We have adapted our exciting costing tool to capture all costs and resource use by line items/input (CAPEX, materials, joint/overhead, etc.) and activity (including development, setting up and routine implementation activities). We hope that the detailed cost analysis will provide useful information to estimate potential cost at scale for Neotree.

Reviewer: 2

Comments to the Author:

1. Background characteristics of newborns admitted to the hospitals might be different by time and location, especially if the interventions, directly or indirectly, affect awareness of families to bring newborns sooner to the hospitals. I think it is important to clarify whether there is any measure to assess characteristics of the input neonates to the system?

-Neotree is replacing paper-based data capture at the three intervention sites. Data on background characteristics of each admitted neonate and their mother is determined in each clinical setting. Typically clinical teams collect data on mothers age, marital status, area of residence, and clinical details pertaining to the birth.

-Neotree supports data capture and clinical care within neonatal intensive care units. It does not perform any health promotion to raise awareness among mothers/communities

-Please note that we are not attempting to evaluate the effectiveness of the intervention at this point. We are still in a pilot stage. Data from comparator sites are being collected only to inform sample size calculations for a future large scale roll out and evaluation. We therefore acknowledge that the background characteristics of patients will vary by hospital.

2. Please provide data on baseline case fatality for the control hospitals

-Currently we do not know the case fatality rates at comparator hospitals. We aim to collect six months of data that will allow us to determine this. These estimates will inform a sample size calculation for large scale roll out and evaluation.

3. Please provide the link to access open-source code on GitHub

-Thank you. We have added this on p9. <https://github.com/Neotree/Neotree>

4. Please clarify whether a linkage is available between the Neotree data and Malawi health system (similar to the linkage to the Zimbabwean health system)

-Not at the time of devising the protocol. We will work with our Malawian colleagues in the Ministry of Health to see if data linkage can be piloted once an electronic health care record system has been developed ready for implementation in Malawi. We will also pilot linkage of Neotree with DHISv2 the most commonly used aggregate data system for reporting of health service data in the African region. (p10)

5. The researchers have considered some interviews with parents/carers; are they involved in any of the processes which are related to Neotree (such as data collection, visualization or export of data)?

-No. Typically mothers accompanying sick neonates, are themselves recovering from medical interventions. Mothers or their relatives provide verbal answers to questions asked by healthcare professionals who enter their answers into the app.

6. If the process of Neotree development is not repeated in each new setting, then its introduction costs to new hospitals will be different from the first few pilot hospitals. I think this should be clarified in the costing method, or should be considered as one of the limitations of costing

-We expect very small development costs when Neotree is introduced in new hospitals, and we agree with the reviewer that the costs (including setting up and implementation costs) are not necessarily generalisable to other hospitals within Malawi and Zimbabwe, let alone other similar settings. We will try to reflect all these uncertainties by running few scenarios. We have now acknowledged these in the discussion section (p16).

7. The supplementary tables have been slitted in several pages, and it's hard to follow the contents. I suggest to provide an excel version of facilitate reviewing and working with them.

- Thank you. We uploaded excel files in the original submission. These split when the pdf proof was generated. Hopefully we can resolve this at the typesetting stage.